# Fault Current Limitation in Electrical Power Networks Containing HTS Cable and HTS Fuse

**DOI:** 10.3390/ma15248754

**Published:** 2022-12-08

**Authors:** Pavel N. Degtyarenko, Vladimir V. Zheltov, Nikolay N. Balashov, Andrey Yu. Arkhangelsky, Alena Yu. Degtyarenko, Konstantin L. Kovalev

**Affiliations:** 1Joint Institute for High Temperature of the Russian Academy of Sciences, Moscow 125412, Russia; 2S-Innovations LLC, Moscow 117246, Russia; 3P.N. Lebedev Physical Institute of the Russian Academy of Sciences, Moscow 119991, Russia; 4Moscow Aviation Institute, National Research University, Moscow 125993, Russia

**Keywords:** 2G HTS wires, HTS cables, short circuit, HTS Fuse

## Abstract

Numerical calculations of parameters of an electrical power network where an HTS fuse is used as a fault current limiting device have been done. The calculations were performed for networks containing different types of HTS cables as well. The design of HTS fuse was developed based on the numerical calculation for the network-rated parameters considering the special types 2G HTS tape characteristics. The distinctive feature of these tapes is the minimal thickness (about 30 µm) of the substrate at the critical current 450–600 A. The tests were performed at a voltage of 1 kV and demonstrated the ability of circuit breaking at fault currents about 3–4 kA. A comparison of experimental results with the calculations allows us to conclude that the HTS fuse of this design can operate as a fault current limiting device in electrical power networks at various rated voltage levels.

## 1. Introduction

An inevitable consequence of the enhancement of electric power systems and the growth of their power is the subsequent increase of fault (short circuit) currents. Nowadays, the fault current level has become a critical parameter limiting the further development of electric power systems. A strategic way to solve this problem consists of mounting and using various types of fault current limiters, and the most promising of the latter are those having high-temperature superconducting (HTS) elements [1]. Superconducting fault current limiters (SFCL) differ from similar conventional devices by the low level of power losses and having an instantaneous reaction to excess current, including fault current as well. In the normal operating modes, these devices do not actually affect the system operation, but if a fault event occurs, it introduces into the network an impedance necessary to limit fault currents to an acceptable level. At the present time, a lot of companies in various countries of the world have developed and built SFCL, e.g., AMSC (Boston, MA, USA) and Siemens (Calais, France)—the first device of 115 kV/1.2 kA [2,3], KEPCO (Naju-si, South Korea)—154 kV [4,5], SuperOx (Moscow, Russia)—220 kV/1.2 kA [6], Shanghai Jiatong University (Shanghai, China)—10 kV/200 A [7,8]. Using 2G HTS tape produced by Shanghai Superconductor Technology Co., Ltd. (SST), (Shanghai, China), the company Zhongtian Technology (Nantong, China) developed a 220 kV/1.5 kA SFCL [9], and Guangdong Power Grid (Guangzhou, China)—a device of ±160 kV/1.0 kA [10]. However, the costs of various fault current limiters (including SFCL as well) are ca. 10,000 $ per 1 MVA of rated power, i.e., the price of a high voltage device can reach 10 million $ per unit.

Another possible way of fault current limitation in various electric circuits is the use of an HTS fuse (HTSF) [11,12]. This device is a kit of 2G HTS tapes placed in a replaceable capsule. Unlike SFCL, it is cheaper, easy to operate and can be rapidly replaced after actuation (the fuse burnout time is 1.5–3 ms, the value of limited current—3–4 rated amplitudes, time of replacement no longer than 10 min).

Building a device of this kind became possible due to the enhancement of the manufacturing technology of 2G HTS tapes produced by S-innovations LLC and the improvement of their current-carrying capacity as well, e.g., recently the critical current has grown by a significant value from 450 to 600 A with the retaining of uniformity over the whole tape length at the simultaneous decreasing of the overall tape thickness [13,14,15,16]. It should also be noted that now most measurements and investigations can be performed with commercial 2G HTS tapes [17,18,19,20]. At the same time, optimization of the structure to achieve the necessary tape parameters is a priority.

In this paper, we propose the fault current limitation technique with the use of HTSF introduced into an electric power network. The algorithm of calculations is given, and the efficiency of fault current limitation by HTSF is calculated. The results obtained (including preliminary experimental ones given below) may be used for electric power networks of various types and parameters.

## 2. Model Circuit

The protection circuit of one of the phases is given in Figure 1. As a circuit part under protection is shown a generator with the HTS cable connected to it in series. The HTSF fuse box can be connected in series before and after the HTS cable. Figure 1 shows the HTSF fuse box connection directly into the HTS transmission line. A non-superconducting shunt, which is an irremovable element of the circuit, is connected in parallel to the fuse box. As a rule, critical currents of the fuse box and HTS cable should exceed the rated current amplitude of the network by 20% since, according to the existing regulations, an overload lower than 20% is not considered to be a fault mode. The peculiarity of the fuse box HTS tapes as compared with ones of the cable consists in having a much lower amount of non-superconducting components. Due to this, heating up and burning out of fuse box HTS tapes in a fault mode take place essentially earlier than the temperature of HTS tapes of cable reaches the critical value.

In the rated operative mode, the HTSF fuse box and HTS cable are in the superconducting state. The total network current is determined by the resistance of the load; *I*_S_ = *I*_F_ and *I*_SH_ = 0. In the schematic diagram of Figure 1, the short-circuit is imitated by closing the short-circuit switch, after which the total network current instantaneously exceeds the fuse box critical current with the subsequent heating up. During the HTSF fuse box destruction, the total network current is displaced into the non-superconducting shunt resistance, which limits the current until the standard circuit breaker actuates.

## 3. Algorithm of Calculations

When performing the numerical simulation of the short-circuit mode, the following characteristics are determined: the surge current (*I*_SC_), the extreme value of the current in the transient process of the short circuit mode—*I*_0_(τ_0_), maximum voltage across the HTSF fuse box, the variation of the temperature and resistance of the latter right up to its destruction, the maximum temperature of cable heating.

As a rule, the surge current is reached within the first quarter of the current variation cycle after the short-circuit start and, due to its short duration, does not actually affect the heating of the network elements. However, for devices operation, which is based on the interaction with the magnetic field, its high values are dangerous since they cause dynamic shocks, which for large devices can amount to hundreds of tons. Depending on the specific situation, the values of *I*_SC_ no higher 4–15 current rms values are allowable.

High values of *I*_0_(τ_0_) at the end of the short-circuit mode are unacceptable since they can cause the destruction of circuit breaker contacts. 

HTS cable heating should not cause the deterioration of its superconducting properties. Generally, 2G HTS tapes are considered to withstand heating up to 200 °C. Decreasing the cable temperature to values lower than the latter is very desirable since it reduces the cable cooling down time necessary for recovering the superconducting state and, subsequently, diminishes the liquid nitrogen consumption.

In the rated operative mode, the resistances of the HTS cable and fuse box are assumed to be zero, and the voltage and current variations are calculated analytically using the following system of equations:(1)$$U=U_{0}\sin\left(\omega t+\varphi \right)\;\;I=\frac{U_{0}}{Z}\sin\left(\omega t\right)\;\varphi =arctg\frac{R_{G}+R_{L}}{\omega \left(L_{G}+L_{L}+L_{F}+L_{S}\right)}\;\;Z=\sqrt{{\left(R_{G}+R_{L}\right)}^{2}+\omega^{2}{\left(L_{G}+L_{L}+L_{F}+L_{S}\right)}^{2}}$$

The moment of occurrence of a short circuit *t*_sc_ is set in the source data of the program in fractions of the current change period. At *t* > *t*_sc_ the current distribution in the network and the voltage variation across the fuse box *U**_F_*(*t*) are determined by the following differential equations:(2)$$\left(L_{G}+L_{S}\right)\frac{dI_{S}}{dt}+I_{S}R_{G}+\left(I_{S}-I_{Sk}\left(T_{k}\right)\right)R_{Ek}\left(T_{k}\right)+U_{F}\left(t,T_{F}\right)=U\left(t\right)\;U_{F}\left(t,T_{F}\right)=\left(I_{S}-I_{CF}\left(T_{F}\right)\right)R_{EF}\left(T_{F}\right)L_{F}\frac{dI_{F}}{dt}\;U_{F}\left(t,T_{F}\right)=R_{SH}I_{SH}+L_{SH}\frac{dI_{SH}}{dt}\;I_{S}=I_{sh}+I_{F}$$
where *I*_Ck_ (TC) and *I*_CF_ (TF) are the critical currents of the superconducting elements of the cable and fuse box depending on their temperatures TC and TF; REC (TC) and REF (TF) are the equivalent resistances of all the non-superconducting components cable and fuse box. The temperature dependence of critical currents of the cable and fuse box *I**_c_* (*T*) is approximated by the following formula [20]:(3)$$I_{c}\left(T\right)=5.3I_{c}\left(T=77K\right)\ln\left(\frac{170-T}{77}\right)$$

The increment of temperatures at each integration step is calculated from the equation:(4)$$\frac{dT}{dt}=\frac{\sum_{i=1}^{N}R_{i}\left(T\right)I_{i}^{2}\left(T\right)}{\sum_{i=1}^{N}C_{i}\left(T\right)}$$
where *R*_i_ (*T*), *I*_i_ (*t*, *T*) and *C*_i_ (*T*) are the resistance, current and heat capacity of component i of the cable or fuse box, and the summation is carried out for all components. The results of the calculations are presented in our previous work [21].

When calculating the fuse box destruction processes, the relations describing the entire range of changes in the aggregate states of the components are used. The block diagram of the algorithm is given in Figure 2.

## 4. Results of Calculation

Previously we investigated an opportunity of fault current limitation by the introduction of the active electric resistance of HTS power transmission lines appearing due to the quench of their HTS components to a normal state [21]. To evaluate the effectiveness of the additional connection of a superconducting fuse, it is advisable to compare the calculations of this work with the results of this section.

In [21], the calculations for four cables (Cable 1–Cable 4) were performed, differing in composition and cross-sectional areas of non-superconducting components. Any phase of each cable was assumed to contain 60 HTS tapes manufactured by SuperPower (USA) in standard or modified versions. The tapes were multilayer composites. In the tapes of the standard version [22,23] the order of alternating layers of the composite and their thicknesses were the following: Hastelloy substrate (h)—50 µm; buffer layers with a total thickness of 0.2 µm, superconducting coating (S)—1 µm; silver coating (Ag)—2 µm; stabilizing copper coating (Cu) 40 µm. The width of the tapes was 4 mm. The critical current for all the tape modifications was taken 85 A. Therefore, the phase critical current for all the cables was 5.1 kA. In the design versions of cables Cable 1–Cable 3, the amount of copper has consistently decreased: the cross-section of Cable 1, in addition to the copper coating of the tapes themselves, contained 120 mm^2^ extra copper, in Cable 2 was used only the tapes of the standard version, in Cable 3—only the tapes without copper coating. In Cable 4 we considered the use of the tapes of the hypothetic design version, i.e., also without copper coating and with the doubled substrate thickness of 100 µm.

When performing the calculations, it was determined the critical length of the HTS power transmission line was, i.e., that one at which the cable temperature within the short-circuit time τ_0_ = 0.375 s reached the value whereat HTS tapes lose their superconducting properties (*T*_m_ = 473 K). Generally, if a short-circuit occurs, a regulation is used that requires, after the circuit breaker actuation, performing two additional attempts to connect the load to the network with a time interval of no more than 2 s. That is why assuming the time intervals between connections not to be sufficient for the cable cooling down there had been taken a value of τ_0_ = 0.375 s corresponding to the triple time of actuation of standard circuit breakers (0.125 s). 

For the purpose of comparative evaluation of the efficiency of HTSF use, we presented the results of short-circuit modes calculations for cables of the same design and composition as in [21]. In our case, we change the connection to the network according to the schematic diagram of Figure 1. However, the length of all the cable types is taken one and the same *l*_C_ = 50 m (apparently, it is the minimum value of length being of practical interest). Furthermore, due to the specifics of processing the short-circuit mode in the presence of HTSF, we accepted the final fixation time of the cable temperature to be equal to the standard circuit breaker actuation time τ_0_ = 0.125 s.

The HTSF box for one cable phase consists of 20 special HTS tapes developed and manufactured by S-Innovations LLC [23]. These tapes do not have the copper coating. The thicknesses of other components are the following: h (Hastelloy) = 40 µm, S (superconductor) = 2 µm, Ag = 3 µm. Critical current tapes at their width of 4 mm are 250 A. Therefore, the critical current of 20 tapes in the fuse box exceeds that of one of the cables’ phases. The inductances of the fuse box and shunt are *L*_F_ = 0.01 mH and *L*_SH_ = 0.08 mH. The shunt resistance is *R*_SH_ = 15 Ohm. This value is higher than the load resistance, and, that is why, after the fuse box burnout, the current is lower than the rated value.

The time dependences of network and fuse box currents in the short-circuit mode for cables Cable 1 and Cable 2 are shown in Figure 3 and Figure 4. The most essential characteristics of short-circuit modes are presented in Table 1.

All the data given are obtained for short-circuit starting moment *t**_0SC_ = 0.2. This value of *t**_0SC_ is chosen because it corresponds to the worst characteristics of the short-circuit mode.

## 5. Discussion

### 5.1. Analysis of the Results

The comparison of data in [21] for Cable 1 and Cable 2 shows the introduction of extra copper into a cable to be senseless. Apart from the significant critical length increase, its presence causes quite inacceptable values of the surge current. That is why, even at the cable length much more than the critical one, current limiting devices are required. The surge current for Cable 1 and Cable 2 significantly exceeds the ordinary rated values. However, the solution for the current limitation of these cables is not unambiguous. The result depends upon the actual cable length and specific requirements for the surge current limitation for other elements of the network under protection. The HTSF connection to Cable 3 and Cable 4 provides necessary short-circuit mode characteristics that are equivalent to those of circuits using much more expensive current limiting devices, e.g., resistive SFCL [6,24]. At the same time, the results do not depend on the cable length. The radical lowering of the maximum cable temperature should be noted as well. Moreover, already at the start of the circuit breaker actuation, the cable temperature drops up to the values allowing the recovery of the critical current exceeding the amplitude *I*_0_. This means that if it is necessary to carry out the subsequent test inclusions established by the regulations, which must be carried out when the fuse box is destroyed, the cable current becomes totally superconducting.

The HTSF connection to Cable 4 does not actually affect the short-circuit mode characteristics. The only effect is to be able to provide these characteristics when the cable length is significantly less than the critical value. However, it should be noted that the calculation of these design versions of cables was performed in [21] in order to solve the theoretical task, i.e., to determine the limit to which the results tend to decrease the copper coating thickness of HTS tapes. However, manufacturing an HTS cable with tapes completely devoid of copper coating is hardly likely. To ensure a stable operation of such cable, there must be obtained an ideal uniformity of tape characteristics and heat transfer conditions over the whole length with the exclusion of any disturbances. Finally, the possibility of damage to the thin silver coating during the cable manufacturing process should be excluded. When manufacturing a short HTSF fuse box, these conditions can be fulfilled, but they can hardly be ensured over the whole cable length. However, the manufacturing of cables having HTS tapes with reduced copper coating thickness (or using coatings of other materials) seems to be quite realistic.

The length of HTS conductors in the windings of resistive SFCL is generally several kilometers, which cannot be compared at all with the amount of HTS materials required for making an HTSF fuse box. That is why the evaluated cost of the latter, as compared with one of a resistive SFCL, is lower by 1–2 orders of magnitude. The frequency of short-circuit appearance in most electric power plants and substations of high and average power does not exceed 1–2 times per year. Subsequently, the replacement of fuse boxes cannot affect the operating costs. The time required to replace a burnt fuse box should not exceed 10 min at the optimal design of the contact devices. However, there can be situations where, according to the operation conditions, the time of short-circuit processing should be essentially lower. For example, in Russia, there are often regulations requiring two more test switches to be made with an interval of no more than 2 s after circuit breaker actuation caused by a short circuit since there is a possibility of self-elimination of the latter. To fulfill these regulations, there must be a reserve kit of fuse boxes connected only after the short-circuit elimination. Test switching should be made even with the destroyed fuse box, and their result is determined by the value of the steady current. If the short-circuit mode is retained, the network has the current corresponding to the shunt resistance. However, if it is eliminated, there is a much lower current corresponding to the sum of the shunt and load resistances connected in series.

### 5.2. Preliminary Experimental Results

As it was shown before, HTS tapes designed for the reliable operation of the HTSF device should be made without copper coating and, due to obvious reasons, must withstand multiple heating and cooling cycles and retain their superconducting properties. That is why these obligatory requirements should be confirmed experimentally.

To this end, the tapes manufactured by S-Innovations LLC were subjected to preliminary tests. On the one hand, conventional DC low-voltage current-voltage characteristics (CVC) measurements and checking up the current-carrying capacity in the AC mode, and, on the other hand, preliminary high-voltage AC measurements at 1 kV aimed to check up the ability of HTS tapes short samples (ca. 120 mm of length) to operate as HTSF themselves.

In low voltage experiments, we measured critical currents *I*_c_ obtained from the CVC curves in accordance with the generally accepted criterion of the electric field across the sample of 1 µV/cm. To avoid any electromagnetic interference, the measurements were made in the foam plastic cryostat. The sample holder had tubular gas-cooled current leads, with the cross-section area allowing carrying currents up to 1200 A, i.e., more than twice as much as the average critical currents of HTS tapes used. This design makes it possible to prevent unwanted heating of the sample when current flows through it, and that is why any subsequent deterioration of the CVC curve. Together with the use of the screened twisted pair to transmit a useful signal from a sample to the system voltmeter Keithley 2000 this technique allowed us to measure the voltage across the sample at an accuracy no worse than 0.1 µV. It was quite sufficient for the length of the measuring section of 10 cm with the appropriate critical current voltage of 10 µV. The sample holder for low-voltage experiments is shown in Figure 5.

The AC measurements at 50 Hz were performed with the use of AC power supply California Instruments 1251 connected with the system of transformers with the output current amplitude up to 1600 A at the total harmonic distortion (THD) coefficient of no more than 2% within the range of currents used.

After obtaining the critical current value in DC measurements, the sample was connected to the AC circuit to check its current-currying capacity in the AC mode up to the amplitudes of 95% of *I*_c_ measured before.

The measurements showed the average critical current values of the HTS tapes 500–550 A to correspond to the ones given by the manufacturer. Subsequently, all the experiments in the AC mode showed the samples to be able to carry the current amplitude specified above.

Additionally, we checked the ability of the tapes to retain their superconducting properties within a long time, from 2 weeks to 2 months after the first testing. Ca. 75% of samples were able to maintain the critical current value after the specified exposure time, and the difference from the values obtained in the first tests did not exceed 0.1%.

After the selection of appropriate samples, we tested them in the same sample holders, which were additionally protected by high-voltage insulation screens in a 1 kV circuit consisting of the power transformer and thyristor switch (they were used instead of the generator in Figure 1) generating short-circuit current amplitudes of 4 kA maximum. The curves *I*(t) obtained showed approximately 60% of these selected samples to be able to operate as fuses with the actuation time of 1–2 ms (compared with the rated actuation time of ordinary electromagnetic circuit breakers of 120 ms) and without the formation of an electric arc, which was observed only for samples with the copper coating. As a load in the normal operating mode before the short-circuit, we used high-current resistors utilized in braking circuits of electric locomotives.

Therefore, these preliminary results allow us to hope for further successful use of tapes as HTSF. Certainly, these experiments will be continued at both 1 kV and higher voltages with the already made new larger sample holders able to operate at voltages up to 35 kV and then described in our further papers in much more detail.

## 6. Conclusions

With the use of the numerical simulation method, it has been investigated the efficiency of short-circuit current limitation performed by the HTSF introduced into the electric power network having an HTS transmission line. The efficiency of the network protection, including the HTS cable protection as well is shown not to be inferior to the efficiency of using known fault current limiters. At the same time, the HTSF evaluated cost is at least by order of magnitude lower than one of any other HTS fault current limiter design. The only disadvantage of HTSF is the necessity of replacement of the destroyed fuse box, which can require up to 10 min of time. However, this time can be significantly reduced by using the reserved HTSF connected directly after the short-circuit elimination. Preliminary experimental results given above showed the principal opportunity of using HTS tapes as HTSF. This opportunity is confirmed by the high-current carrying capacity of the S-Innovations 2G HTS tapes without copper coating based on different important factors for this application. The former is the ability to burn without arcing. The latter is the ability to retain superconducting properties after long exposure in the open air for as long as 2–8 weeks and even more. This allows us to hope for further use of these experimental results in HTSF operating at higher voltages and loads that will be made in our future papers.

## Figures and Tables

**Figure 1 materials-15-08754-f001:**
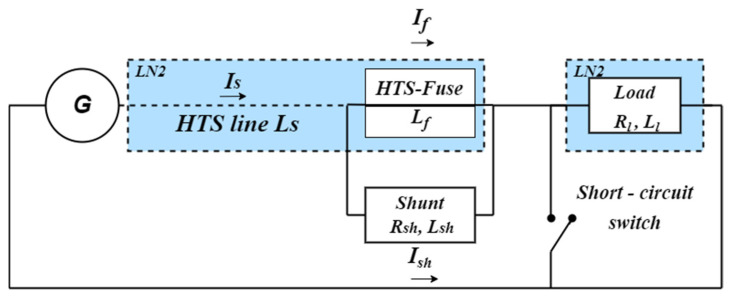
Schematic diagram of the single-phase short-circuits imitation.: *U*_G_, *R*_G_, *L*_G_ are, respectively, the voltage, active resistance and inductance of the generator, *R*_L_, *L*_L_—are, respectively, the active resistance and inductance of the load, *I*_S_ is the total network current, *L*_F_, *I*_F_ are, respectively, the inductance and current of the fuse box, *R*_SH_, *L*_SH_ and *I*_SH_ are, respectively, the active resistance and inductance of the shunt, *L*_C_ is the cable inductance.

**Figure 2 materials-15-08754-f002:**
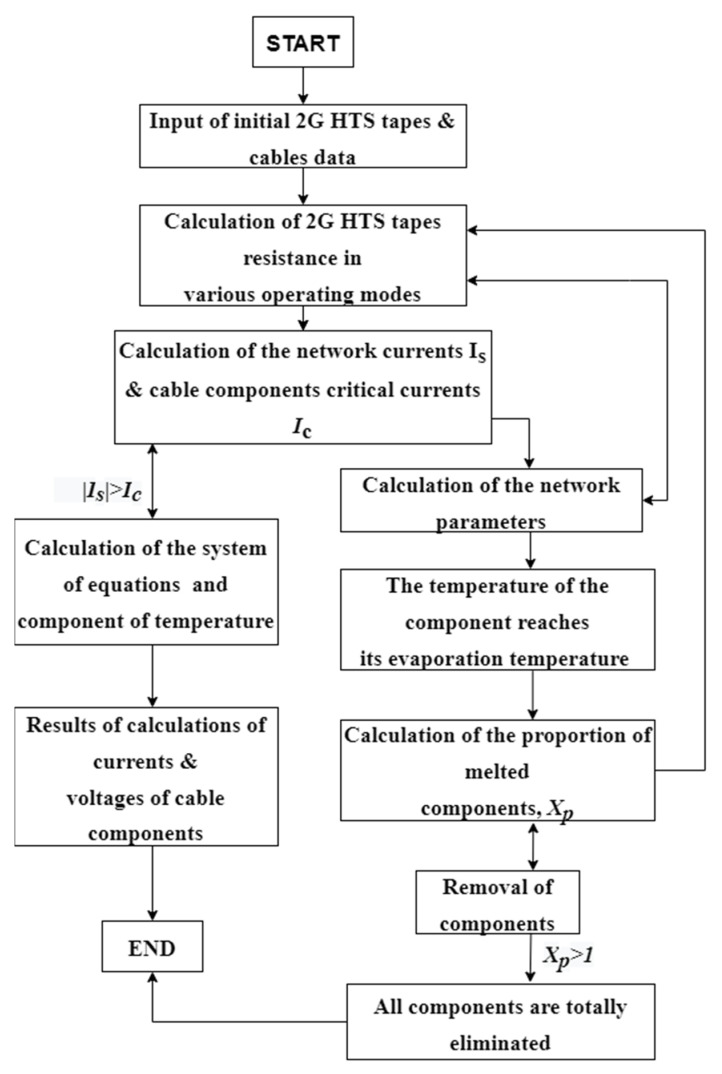
The block diagram of the calculations algorithm.

**Figure 3 materials-15-08754-f003:**
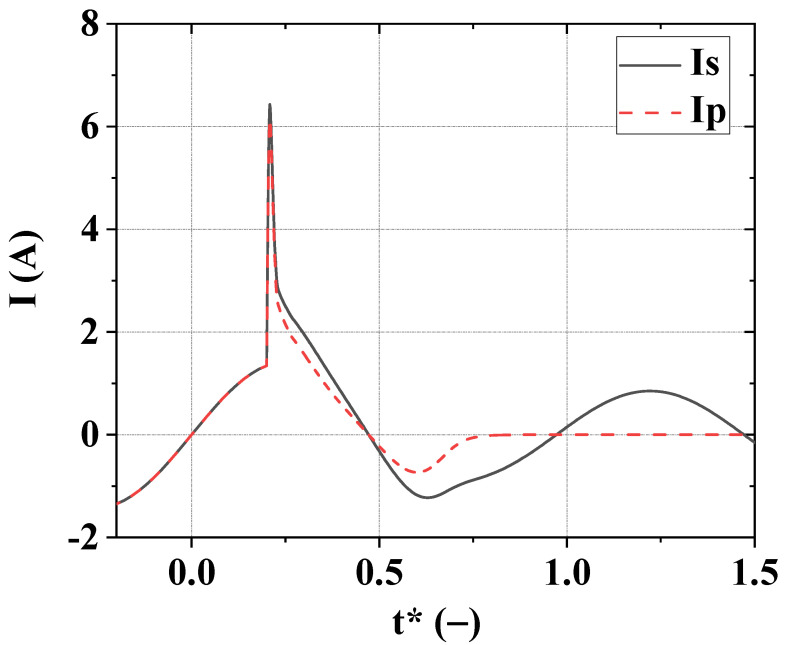
Variation of dimensionless currents of the network *I**_S_ = *I*_S_/*I*_L_ (solid line) and fuse box *I**_F_ = *I*_F_/*I*_L_ (dashed line) for Cable 1. The dimensionless time is *t** = *t f*, *f* = 50 Hz, *t**_0SC_ = 0.2.

**Figure 4 materials-15-08754-f004:**
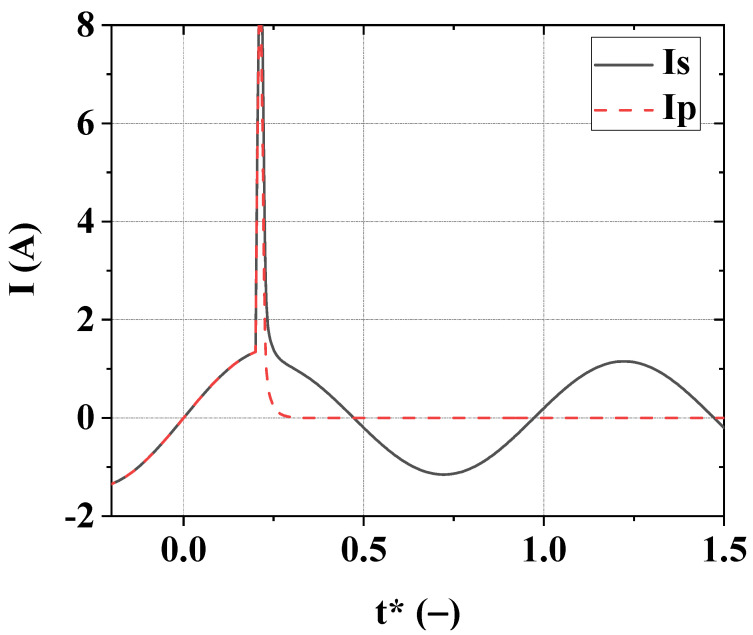
Variation of dimensionless currents of the network *I**_S_ = *I*_S_/*I*_L_ (solid line) and fuse box *I**_F_ = *I*_F_/*I*_L_ (dashed line) for Cable 2. The dimensionless time is *t** = *t* · *f*, *f* = 50 Hz, *t**_0SC_ = 0.2.

**Figure 5 materials-15-08754-f005:**
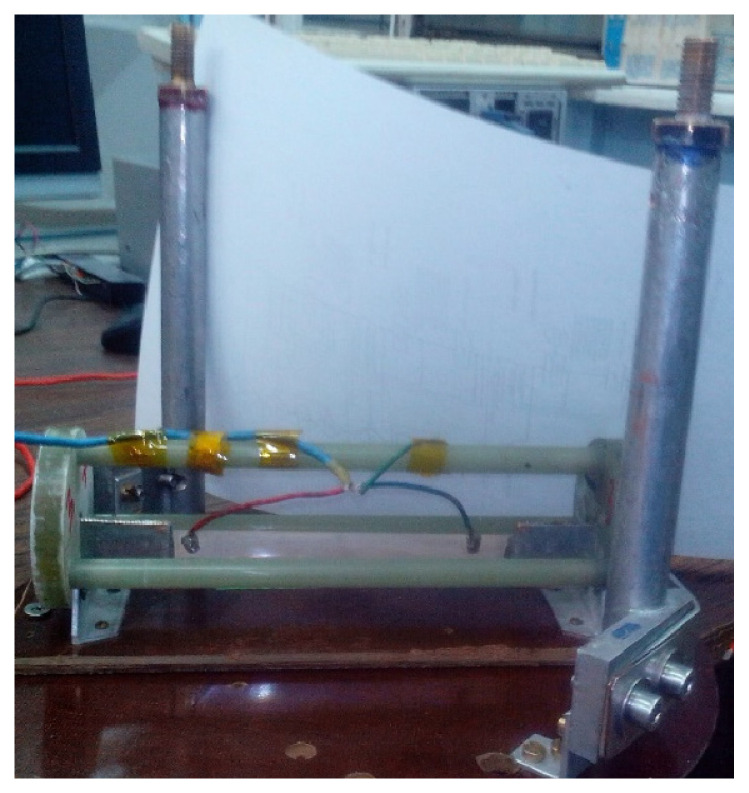
Sample holder with the S-Innovations 12 mm 2G HTS tape 120 mm in length without copper coating prepared for the low voltage check-up experiments. Prior to the high voltage experiments, the potential taps were removed, and the current leads were additionally insulated.

**Table 1 materials-15-08754-t001:** Results of calculations of the short-circuit mode characteristics in HTS power transmission lines with HTSF connected according to the schematic diagram of Figure 1. The initial data for calculations are the same as in Table 1. The cable length is 50 m. The designations are the following: *I**_SC_ = *I*_SC_/*I*_L_ is the dimensionless surge current; *I**_0_ = *I*_0_/*I*_L_ is the amplitude of the current established after the destruction of the fuse box HTS, Δ*t* is the time interval from short-circuit starting to the moment of the fuse box, destruction *T*_CM_ is the maximum cable temperature, *T*_C_(τ_0_) is the cable temperature to the moment of the circuit breaker actuation (τ_0_ = 0.125 s).

Type of Cable	*I**_SC_	*I**_0_	Δ*t* [ms]	*T*_CM_ K	*T*_C_(τ_0_) K
Cable 1	10.03	1.15	2.55	79.3	79.2
Cable 2	10.00	1.15	2.66	97.0	79.1
Cable 3	6.43	0.85	13.6	465	465
Cable 4	7.19	0.99	5.42	427	427

## Data Availability

The data that supports the findings in this study are available from the corresponding author upon reasonable request.

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
