# Peer review of "Fault Current Limitation in Electrical Power Networks Containing HTS Cable and HTS Fuse"

_materials, 2022, doi:10.3390/ma15248754_

Round 1

Reviewer 1 Report

Please, use dot as decimal character. (text and pictures)

I think that one or two pictures of the measuring system would significantly complement the article.

Author Response

First of all, we would like to thank for the careful reading of our work.

  1. Please, use dot as decimal character. (text and pictures).

We have checked and revised it. All the commas in decimal characters are replaced by dots.

  1. I think that one or two pictures of the measuring system would significantly complement the article.

Dear referee, in order not to increase the volume of the article, we specifically agreed that our experimental results are preliminary. We emphasized that check-up low voltage experiments were made with the use of conventional measurement circuits which we would like to show in our further articles. As to the high-voltage experiment, i.e., testing of the HTSF itself, to save the volume of the paper, we have added in the text an explanatory sentence.

Here it is:

The high-voltage circuit to test the HTSF itself was almost the same as shown in Figure. 1. But, instead of the generator, we used a high-power transformer with the output voltage of 1 kV and the thyristor switch, with which this voltage was supplied to the circuit. Of course, we did not have any HTS cables in the experimental circuit, since our task consisted in testing the HTSF itself.

Reviewer 2 Report

The paper titled “Fault current limitation in electrical power networks containing 2 HTS cable and HTS fuse”, investigated the fault current limitation response of HTS fuse which is connected with HTS cable in series in a power network. The topic is certainly worthwhile for a sound study. The paper is well written and organized.

Authors are suggested to address the following comments to improve the article in the revised version.

1.     HTS fuse is functioning to limit the fault current and also break the circuit. However, have you considered the over-current scenarios, where the current is higher than the rated current and the critical current but not significantly large? It will be great to show some results on this aspect by simulation.

2.     It lacks a schematic of the HTS fuse designed for the system. Will you add one figure to present this for the sake of the readers, and together with a list of table for its specifications.

3.     Conclusion needs further improvement to highlight the main outcomes and summary out the study. The present conclusion is so short.

Author Response

First of all, we would like to thank for the careful reading of the article.

  1. HTS fuse is functioning to limit the fault current and also break the circuit. However, have you considered the over-current scenarios, where the current is higher than the rated current and the critical current but not significantly large? It will be great to show some results on this aspect by simulation.

We are very sorry, but we considered only the situation with the maximum fault current for our test circuit without adding any adjusting elements. Our immediate task was to show the fundamental possibility of the operation of the HTS tape as a current-carrying fuse. However, we accept this remark and hope to consider the scenario proposed in our further papers.

  1. It lacks a schematic of the HTS fuse designed for the system. Will you add one figure to present this for the sake of the readers, and together with a list of tables for its specifications.

We added a photo of the sample holder. Among various HTS conductors we have chosen that one with the HTS tape which proved to be optimal for our purposes. It doesn’t have the copper coating and burns without arcing. This tape made by S-Innovations LLC has the following characteristics: width – 12 mm, HTS layer thickness – 1 μm, Hastelloy substrate thickness - 30 µm, average Ic=520 A, the length of the tape current-carrying elements is 120 mm.

  1. Conclusion needs further improvement to highlight the main outcomes and summary out the study. The present conclusion is so short.

We completed the conclusions according to your recommendations.

Reviewer 3 Report

The article conducts a numerical analysis of the performance of a fuse made of HTS 2G tapes. The analysis was performed with the cooperation of the fuse with a superconducting line. I have several questions for the content.

1. why was this configuration chosen? Couldn't the fuse itself be tested?

2. is it reasonable to use this type of fuse to protect transformers?

3. what is new in the work? That the HTS 2G tape will burn out under certain specified conditions is common knowledge?

Notes on the text.

1. why are different cable lengths given in Table 1? If the calculations were theoretical and had no reference to real objects then only calculations for the same length of cables are justified.

2. The asterisk used in the formulas is not a multiplication symbol.

3. SuperPower introduces designations for manufactured cables. In Table 1 and at the beginning of Chapter 4, cable designations should be given.

4. Chapter 4 describes the construction of individual cables manufactured by SuperPower. Presenting this construction in the form of illustrative drawings would increase the readability of the description. The same applies to the products of the S-Innovations company.

5. in the text in several places appear decimal fractions with a comma should be corrected to a period. This remark also applies to drawings.

6. In figure 2 and 3 the timeline is without units. I don't really understand why. I also do not understand the numerical values on the timeline. How to interpret the values 0.5 1.0 1.5? You can't give the time directly in milliseconds and in the text state that the analysis was done at a grid frequency of 50 Hz? Admittedly, in the caption it is given how to recalculate the values from the timeline, but why complicate life for readers?

7. what are Js and Jp in Figures 2 and 3?

8. The paper lacks a description of the calculation algorithm. There should be a block diagram showing the calculation algorithm.

9. The paper lacks verification of calculations with data obtained from the experiment on real objects, thus it is difficult to verify the validity of the results obtained. To increase the credibility of the article, an experiment on one of the objects mentioned would suffice.

10. literature items should be formatted correctly. In the current form it is done incorrectly.

Author Response

First of all, we would like to thank for the careful reading of the article. The obtained comments give us a chance to improve our work.

  1. Why was this configuration chosen? Couldn't the fuse itself be tested?

This configuration was chosen since it is the simplest one, and in our preliminary experiments, we tested the HTSF itself as it is written in the Chapter “Preliminary experiments” but without a HTS cable, which is present only in our theoretical calculations and considerations.

  1. Is it reasonable to use this type of fuse to protect transformers?

We think it is, but in our first experiments we used an active load only. As we answered Reviewer 2, our immediate task was to show the fundamental possibility of the operation of the HTS tape as a current-carrying fuse. An opportunity of using the HTSF for transformers protection could be checked up in our further experiments with new sample holders what we hope to report in our further papers. But here, we think, active load under protection would be sufficient

  1. What is new in the work? That the HTS 2G tape will burn out under certain specified conditions is common knowledge?

Certainly, not only 2G HTS tapes but any conductor will burn under specified conditions. But the point is that not all the conductor burning conditions are suitable for fuse operation. However, S-Innovations LLC has manufactured a very thin tape design of which is suitable for this and we have confirmed it in our preliminary experiments. At least when protecting an active load it burns without arcing and sample holder damage.

Remarks:

  1. Why are different cable lengths given in Table 1? If the calculations were theoretical and had no reference to real objects then only calculations for the same length of cables are justified?

Of course, in Table 1 taken from our previous paper [22] are given theoretical results only since we do not have real HTS cables at our disposal. But the different lengths of the cables design of which is briefly described in the first column of Table 1 are caused by the fact that there should be found the length at which a HTS cable itself could operate as current limiting device and this length is different even for one and the same cable design. But the results of calculations show that even at optimal cable lengths marked by *, the cables can heat up to high temperatures if they are not protected by a HTSF.

  1. The asterisk used in the formulas is not a multiplication symbol.

We have checked and revised it. All the formulae have a proper multiplication symbol now.

  1. SuperPower introduces designations for manufactured cables. In Table 1 and at the beginning of Chapter 4, cable designations should be given.

There are only our models, which we built for our calculations of SuperPower tapes rather than real cable designs manufactured by this company. That is why there are only brief design descriptions given in the first column of Table 1.

  1. Chapter 4 describes the construction of individual cables manufactured by SuperPower. Presenting this construction in the form of illustrative drawings would increase the readability of the description. The same applies to the products of the S-Innovations company.

There are only models that we have already written about in the previous paragraph of our answer.

  1. In the text in several places appear decimal fractions with a comma should be corrected to a period. This remark also applies to drawings.

We have checked and revised it. All the commas in decimal characters are replaced by dots.

  1. In figure 2 and 3 the timeline is without units. I don't really understand why. I also do not understand the numerical values on the timeline. How to interpret the values 0.5 1.0 1.5? You can't give the time directly in milliseconds and in the text state that the analysis was done at a grid frequency of 50 Hz? Admittedly, in the caption it is given how to recalculate the values from the timeline, but why complicate life for readers?

The timeline in Figures 2 and 3 is given in relative units - fractions of grid period since the latter are used in our program for calculations and this is described in the figure captions. The grid frequency in all our calculations and preliminary experiments is one and the same - 50 Hz. But if we take another frequency, e. g. 60 Hz we shall have the same pattern, which seems more obvious to us than if we used absolute milliseconds. Therefore, we would like to request for retaining the existing axis title.

  1. What are Js and Jp in Figures 2 and 3?

We renamed these values as Is and Ip on Figure 2 and 3, but these Figures in new version of the manuscript are renumbered as Figure 3 and 4, respectively. The former is the load current; the latter is the HTSF current.

  1. The paper lacks a description of the calculation algorithm. There should be a block diagram showing the calculation algorithm.

We added simplified block diagram showing this algorithm in Figure 5.

  1. The paper lacks verification of calculations with data obtained from the experiment on real objects, thus it is difficult to verify the validity of the results obtained. To increase the credibility of the article, an experiment on one of the objects mentioned would suffice.

We are very sorry, but the direct verification of our calculations could not be performed since we do not have any HTS cables at our disposal as it was written above. The only thing we could do is the experimental checking up of the ability of HTS tapes of the design chosen and manufactured by S-Innovations LLC to operate as HTSF themselves. We emphasize that our only preliminary experiments have shown that this is possible. The experiments with more complicated objects than high-power railway resistors we used will be described in our further papers as we hope. But just such cables which we have chosen as calculation models, are not manufactured yet.

  1. Literature items should be formatted correctly. In the current form it is done incorrectly.

We have checked and revised all. The references are given in a proper format.